# FusionVision: A Comprehensive Approach of 3D Object Reconstruction and Segmentation from RGB-D Cameras Using YOLO and Fast Segment Anything

**DOI:** 10.3390/s24092889

**Published:** 2024-04-30

**Authors:** Safouane El Ghazouali, Youssef Mhirit, Ali Oukhrid, Umberto Michelucci, Hichem Nouira

**Affiliations:** 1TOELT LLC, AI Lab, 8406 Winterthur, Switzerland; umberto.michelucci@toelt.ai; 2Independent Researcher, 75000 Paris, France; 3Independent Researcher, 2502 Biel/Bienne, Switzerland; 4LNE Laboratoire National de Metrologie et d’Essaies, 75015 Paris, France; hichem.nouira@lne.fr

**Keywords:** RGB-D, 3D reconstruction, point-cloud, SAM, 3D object detection, 3D localization

## Abstract

In the realm of computer vision, the integration of advanced techniques into the pre-processing of RGB-D camera inputs poses a significant challenge, given the inherent complexities arising from diverse environmental conditions and varying object appearances. Therefore, this paper introduces FusionVision, an exhaustive pipeline adapted for the robust 3D segmentation of objects in RGB-D imagery. Traditional computer vision systems face limitations in simultaneously capturing precise object boundaries and achieving high-precision object detection on depth maps, as they are mainly proposed for RGB cameras. To address this challenge, FusionVision adopts an integrated approach by merging state-of-the-art object detection techniques, with advanced instance segmentation methods. The integration of these components enables a holistic (unified analysis of information obtained from both color *RGB* and depth *D* channels) interpretation of RGB-D data, facilitating the extraction of comprehensive and accurate object information in order to improve post-processes such as object 6D pose estimation, Simultanious Localization and Mapping (SLAM) operations, accurate 3D dataset extraction, etc. The proposed FusionVision pipeline employs YOLO for identifying objects within the RGB image domain. Subsequently, FastSAM, an innovative semantic segmentation model, is applied to delineate object boundaries, yielding refined segmentation masks. The synergy between these components and their integration into 3D scene understanding ensures a cohesive fusion of object detection and segmentation, enhancing overall precision in 3D object segmentation.

## 1. Introduction

The significance of point-cloud processing has surged across various domains, such as robotics [1,2], medical field [3,4], autonomous driving [5,6], metrology [7,8,9], etc. Over the past few years, advancements in vision sensors have led to remarkable improvements, enabling these sensors to provide real-time 3D measurements of the surroundings while maintaining decent accuracy [10,11]. Consequently, point-cloud processing forms an essential pivot of numerous application by facilitating robust object detection, segmentation, and classification operations.

Within the field of computer vision, two extensively researched pillars stand prominent: object detection and object segmentation. These subfields have captivated the research community for the past decades, helping computers understand and interact with visual data [12,13,14]. Object detection involves identifying and localizing one or multiple objects in an image or a video stream, often employing advanced deep learning techniques, such as Convolutional Neural Networks (CNNs) [15] and Region-based CNNs (R-CNNs) [16]. The pursuit of real-time performance has led to the development of more efficient models, such as Single Shot MultiBox Detector (SSD) [17] and You Only Look Once (YOLO) [18], which demonstated a balanced performance between accuracy and speed. On the other hand, object segmentation goes beyond the detection process allowing delineating the precise boundaries of each identified object [19]. The segmentation process enables a finer understanding of the visual scene and a precise object localization in the given image. In the literature, two segmentation types are differentiated: semantic segmentation assigns a class label to each pixel [20], while instance segmentation distinguishes between individual instances of the same class [21].

One of the most popular object detection models is (YOLO). The latest known version of YOLO is YOLOv8, which is a real-time object detection system that uses a single neural network to predict bounding boxes and class probabilities simultaneously [22,23]. It is designed to be fast and accurate, making it suitable for applications such as autonomous vehicles and security systems. YOLO works by dividing the input image into a grid of cells, where each one predicts a fixed number of bounding boxes, which are then filtered using a defined confidence threshold. The remaining bounding boxes are then resized and repositioned to fit the object they are predicting. The final step is to perform non-maximum suppression [24] on the remaining bounding boxes to remove overlapping predictions. The loss function used by YOLO is a combination of two terms: the localization loss and the confidence loss. The localization loss measures the difference between the predicted bounding box coordinates and the ground truth coordinates, while the confidence loss measures the difference between the predicted class probability and the ground truth class.

SAM [25], on the other hand, is a recent popular deep learning model for image segmentation tasks. It is based on the U-Net architecture commonly selected for medical applications [26,27,28]. U-Net is a CNN that is specifically designed for image segmentation, which consists of an encoder and a decoder, which are connected by a skip connection [29]. The encoder is responsible for extracting features from the input image, while the decoder handles the generation of the segmentation mask. The skip connection allows the model to use the features learned by the encoder at different levels of abstraction, which helps in generating more accurate segmentation masks. SAM gained its popularity because it achieves state-of-the-art performance on various image segmentation benchmarks and many fields, such as the medical field [30], and additional known dataset such as the PASCAL VOC 2012 [31]. It is particularly effective in segmenting complex objects, such as buildings, roads, and vehicles, which are common in urban environments. The model’s ability to generalize across different datasets and tasks has highly contributed to its popularity.

The use of YOLO and SAM is still extensively studied and field-applied by the scientific community for 2D computer vision tasks [32,33,34]. However, in this paper, we focus the study on the involvement possibility of both state-of-the-art algorithms on RGB-D images. RGB-D cameras are depth sensing cameras that capture both RGB-channel (Red, Green, Blue) and D-map (depth information) of a scene (example shown in Figure 1). These cameras use infrared (IR) projectors and sensors to measure the distance of objects from the camera, providing an additional depth dimension to the RGB image with sufficient accuracy. For example, according to F. Pan et al. [35], an estimated accuracy of 0.61 ± 0.42 mm has been assessed on RGB-D camera for facial scanning. Compared to traditional RGB cameras, RGB-D cameras offer several advantages, including:(1)Improved object detection and tracking [36]: The depth information provided by RGB-D cameras allows for more accurate object detection and tracking, even in complex environments with occlusions and varying lighting conditions.(2)3D reconstruction [37,38]: RGB-D cameras can be used to create 3D models of objects and environments, enabling applications such as augmented reality (AR) and virtual reality (VR).(3)Human–computer interaction [39,40]: The depth information provided by RGB-D cameras can be used to detect and track human movements, allowing more natural and intuitive human–computer interaction.

RGB-D cameras have a wide range of applications, including robotics, computer vision, gaming, and healthcare. In robotics, RGB-D cameras are used for object manipulation [41], navigation [42], and mapping [43]. In computer vision, they are used for 3D reconstruction [37], object recognition, and tracking [44,45]. All those algorithm take advantage of the depth information to work with 3D data instead of images. The point-cloud processing allows additional accuracy for the object tracking leading to improved knowledge about its position, orientation, and dimensions in 3D space. This offer distinct advantages compared to traditional image-based systems. Furthermore, RGB-D technologies are also able to surpass diverse lighting conditions [46] due to the use of IR lighting.

This paper presents a contribution in the fields of RGB-D and object detection and segmentation. The primary contribution lies in the development and application of FusionVision, a method that links models originally proposed for 2D images, with RGB-D types of data. Specifically, two known models have been implemented, validated, and adjusted to work with RGB-D data through the use of both the Depth and RGB channels of an Intel Realsense camera. This combination has led to an enhancement in understanding scenes resulting in 3D object isolation and reconstructions without distortions or noises. Moreover, point-cloud post-processing techniques, including denoising and downsampling, have been integrated to remove anomalies and distortions caused by reflectivity or inaccurate depth measurements, as to improve the real-time performance of the proposed FusionVision pipeline. The code and pre-trained models are publicly available at https://github.com/safouaneelg/FusionVision/ (accessed on 28 February 2024).

The rest of the paper is organized as follows. Despite the uniqueness of the proposed pipeline and the scarcity of methods similar to the one proposed in this paper, few related works are discussed in Section 2. A detailed and comprehensive description of the FusionVision pipeline is given in Section 3, where the processes are discussed step-by-step. Following this, the implementation of the framework and results are presented and discussed in Section 4. Finally, the paper’s findings are summarized in Section 5.

## 2. Related Work

The aforementioned YOLO and SAM models have been mainly proposed for 2D computer vision operations, lacking the adaptability for RGB-D images. The 3D detection and segmentation of the objects is, therefore, beyond their capabilities, leading to a need for 3D object detection methods. Within this context, few methods have been studied for 3D object detection and segmentation from RGB-D Cameras. Tan Z. et al. [47] proposed an improved YOLO (version 3) for 3D object localization. The method aims to achieve real-time high-accuracy 3D object detection from point-clouds using a single RGB-D camera. The authors proposed a network system that combines both 2D and 3D object detection algorithms to improve real-time object detection results and increase speed. The used combination of two state-of-the-art object detection methods are [48] performing object detection from an RGB sensor and Frustum PointNet [49], a real-time method that uses frustum constraints to predict a 3D bounding box of an object. The method framework could be summarized as follows (Figure 2):The system starts by obtaining 3D point-clouds from a single RGB-D camera along with the RGB stream.The 2D object detection algorithm is used to detect and localize objects in the RGB images. This provides useful prior information about the objects, including their position, width, and height.The information from the 2D object detection is then used to generate 3D frustums. A frustum is a pyramid-shaped volume that represents the possible location of an object in 3D space based on its 2D bounding box.The generated frustums are fed into the PointNets algorithm, which performs instance segmentation and predicts the 3D bounding box of each object within the frustum.

By combining the results from both the 2D and 3D object detection algorithms, the system achieves real-time object detection performance, both indoors and outdoors. For the method evaluation, the author stated achieving real-time 3D object detection using an Intel realsense D435i RGB-D camera with the algorithm running on a NVIDIA GTX 1080 ti GPU-based system. However, this proposed method has limitations and is subject to noise usually due to bad estimation of depth and object reflectivity.

## 3. FusionVision Pipeline

The implemented FusionVision pipeline could be summarized in six steps in addition to the first step of data acquisition (Figure 3):

**Data acquisition and Annotation**: This initial phase involves obtaining images suitable for training the object detection model. This image collection can include single- or multi-class scenarios. As part of preparing the acquired data, splitting into separate subsets designated for training and testing purposes is required. If the object of interest is within the 80 classes of Microsoft COCO (Common Objects in Context) dataset [50], this step may be optional, allowing the utilization of existing pre-trained models. Otherwise, if the special object is to be detected, or object shape is uncommon or different from the ones in the datasets, this step is required.**YOLO model training**: Following data acquisition, the YOLO model undergoes training to enhance its ability to detect specific objects. This process involves optimizing the model’s parameters based on the acquired dataset.**Apply model inference**: Upon successful training, the YOLO model is deployed on the live stream of the RGB sensor from the RGB-D camera to detect objects in real-time. This step involves applying the trained model to identify objects within the camera’s field of view.**FastSAM application**: If any object is detected in the RGB stream, the estimated bounding boxes serve as input for the FastSAM algorithm, facilitating the extraction of object masks. This step refines the object segmentation process by leveraging FastSAM’s capabilities.**RGB and Depth matching**: The estimated mask generated from the RGB sensor is aligned with the depth map of the RGB-D camera. This alignment is achieved through the utilization of known intrinsic and extrinsic matrices, enhancing the accuracy of subsequent 3D object localization.**Application of 3D reconstruction from depth map**: Leveraging the aligned mask and depth information, a 3D point-cloud is generated to facilitate the real-time localization and reconstruction of the detected object in three dimensions. This final step results in an isolated representation of the object in the 3D space.

### 3.1. Data Acquisition

For applications requiring the detection of specific objects, the data acquisition consists of collecting a number of images using the camera of the specific object at different angles, positions, and varying lighting conditions. The images need to be annotated afterwards with the bounding boxes corresponding to the location of the object within the image. Several annotators could be used for this step, such as Roboflow [51], LabelImg [52], or VGG Image Annotator [53].

### 3.2. YOLO Training

Training the YOLO model for robust object detection forms a strong backbone of FusionVision pipeline. The acquired data are split into 80% for training and 20% for validation. To further enhance the model’s generalization capabilities, data augmentation techniques were employed by horizontally and vertically flipping images, as well as applying slight angle tilts [54].

In the context of object detection using YOLO, several key loss functions play a pivotal role in training the model to accurately localize and classify objects within an image. The Objectness Loss ($O_{L}$), defined in Equation (Equation 1), employs binary cross-entropy to assess the model’s ability to predict the presence or absence of an object in a given grid cell, where $y_{i}$ represents the ground truth objectness label for a given grid cell in the image. The Classification Loss ($CLS_{L}$), as outlined in Equation (Equation 2), utilizes cross-entropy to penalize errors in predicting the class labels of detected objects across all classes (*C* the class number). To refine the localization accuracy, the Bounding Box Loss ($Bbox_{L}$), described in Equation (Equation 3), leverages mean squared error to measure the disparity between predicted ${\hat{y}}_{i}$ and ground truth $y_{i}$ bounding box coordinates. Moreover, $c_{x}$, $c_{y}$ refer to the center coordinates of the bounding box and *w*, *h* are its width and height. Additionally, the Center Coordinates Loss ($C_{L}$), detailed in Equation (Equation 4), incorporates focal loss, including parameters α and γ, to address the imbalance in predicting the center coordinates of objects. These loss functions collectively guide the optimization process during training, steering the YOLOv8 model towards robust and precise object detection performance across diverse scenarios:(1)$$O_{L}=-(y_{i}\cdot \log\left({\hat{y}}_{i}\right)+\left(1-y_{i}\right)\cdot \log\left(1-{\hat{y}}_{i}\right))$$
(2)$$CLS_{L}=-\sum_{c=1}^{C}y_{i,c}\cdot \log\left({\hat{y}}_{i,c}\right)$$
(3)$$Bbox_{L}=\sum_{p\in \{c_{x},\;c_{y},\;w,\;h\}}{\left(y_{i,p}-{\hat{y}}_{i,p}\right)}^{2}$$
(4)$$C_{L}=-\alpha \cdot {\left(1-{\hat{y}}_{i,\mathrm{center}}\right)}^{\gamma }\cdot y_{i,\mathrm{center}}\cdot \log\left({\hat{y}}_{i,\mathrm{center}}\right)$$

Throughout the training process, images and their corresponding annotations are fed into the YOLO network [22]. The network, in turn, generates predictions for bounding boxes, class probabilities, and confidence scores. These predictions are then compared to the ground-truth data using the aforementioned loss functions. This iterative process progressively improves the model’s object detection accuracy until reaching a minimal value of total loss.

### 3.3. FastSAM Deployment

Once the YOLO model is trained, its bounding boxes serve as input for the subsequent step involving the FastSAM model. When processing the complete image, FastSAM estimate instance segmentation mask for all the viewed objects. Therefore, instead of processing the entire image, the YOLO estimated bounding boxes are used as input information to focus the attention on the relevant region where the object is, significantly reducing computational overhead. Its Transformer-based architecture then delves into this cropped image patch to generate a pixel-wise mask.

### 3.4. RGB and Depth Matching

RGB-D imaging devices typically incorporate an RGB sensor, responsible for capturing traditional 2D color images, and a depth sensor integrating left and right cameras alongside an infrared (IR) projector positioned in the middle. The project IR patterns onto the physical object are distorted by its shape, then get captured by the left and right cameras. Afterwards, the disparity information between corresponding points in the two images is used to estimate the depth of each pixel in the scene. The extracted segments resulting as an output of the FastSAM are represented through binary masks in the RGB channel of the cameras. The identification of the physical object in the DS is carried out by aligning both binary masks and depth frames (Figure 4).

Within this alignment process, the transformation between the coordinate systems of the RGB camera and the depth sensor needs to be estimated either using the calibration process or based on the default factory values. Few calibration techniques can be used for the improvement of the matrices estimation, such as [55,56]. This transformation is represented mathematically in Equation (Equation 5):(5)$$Z_{0}\left[\begin{array}{c} u_{0}\\ v_{0}\\ 1 \end{array}\right]=K_{c}T_{cd}K_{d}^{-1}\left[\begin{array}{c} Z\\ u\\ vs.\\ 1 \end{array}\right]$$
where:$Z_{0},u_{0},v_{0}$ represent the depth value and pixel coordinates in the aligned depth image;Z,u,v is the depth value and pixel coordinates in the original depth image;$K_{c}$ is the RGB camera intrinsic matrices;$K_{d}$ is the DS intrinsic matrices;$T_{cd}$ represents the rigid transformation between RGB and DS.

### 3.5. 3D Reconstruction of the Physical Object

Once FastSAM mask is aligned with the depth map, the identified physical objects could be reconstructed in 3D coordinates, taking into account only the region of interest (ROI). This process involves several key steps, including: (1) downsampling, (2) denoising, and (3) generating the 3D bounding boxes for each identified object in the point-cloud.

The downsampling process is applied to the original point-cloud data, which allows the reduction of computational complexity while retaining essential object information. The selected downsampling technique involves voxelization, where the point-cloud is divided into regular voxel grids, and only one point per voxel is retained [57]. Following downsampling, a denoising procedure based on statistical outliers removal [58] is implemented to enhance the quality of the generated point-cloud. Outliers, which may arise from sensor noise are identified and removed from the point-cloud. Finally, for each physical object detected in the aligned FastSAM mask, a 3D bounding box is generated within the denoised point-cloud. The bounding box generation involves creating a set of lines connecting the min and max coordinates along each axis. This set of lines is aligned with the object’s position in the denoised point-cloud. The resulting bounding box provides a spatial representation of the detected object in 3D.

## 4. Results and Discussion

### 4.1. Setup Configuration

For the experimental study, the proposed framework is tested on the detection of three commonly used physical objects: cup, computer and bottle. The setup configuration that has been used is summarized in Table 1.

### 4.2. Data Acquisition and Annotation

For the data acquisition step, a total of 100 images featuring common objects, namely a cup, computer, and bottle, were captured using the RGB stream of a RealSense camera. The recorded images include several poses of the selected 3D physical objects and lighting conditions, as to ensure robust and comprehensive dataset for the model training. The images were annotated using the Roboflow annotator for the YOLO object detection model. Additionally, data augmentation techniques were then applied to enrich the dataset, involving horizontal and vertical flipping, as well as angle tilting (Figure 5).

### 4.3. YOLO Training and FastSAM Deployment

#### 4.3.1. Model Training and Deployment

The training of object detection has been performed unsing both the acquired and augmented images. Figure 6 summarizes the training results and validation curves including loss functions $CLS_{L}$ and $Bbox_{L}$, the precision and recall, and finally the mAP50 and mAP50-95 metrics. mAP50 measures the average precision of a model across different object classes when the overlap threshold (IoU—Intersection over Union) between predicted and ground truth bounding boxes is set to 50%). mAP50-95 is an extension of mAP50 that considers a range of IoU thresholds from 50% to 95%, providing a more comprehensive evaluation of a model’s performance by considering a broader spectrum of overlap criteria. The YOLO algorithm was trained for 300 epochs using the smallest YOLO model variant *yolov8n*, with a learning rate of 0.01 and the Adam optimizer [59].

For the bbox loss (Figure 6a), the train box loss exhibits continuous reduction, requesting the consideration of additional epoches. In contrast, the validation box loss stabilizes around 0.48 approximately at epoch 200, reaching a plateau thereafter. The class loss curve (Figure 6b) demonstrates a quicker convergence, with both train and validation loss curves decreasing rapidly. Around epoch 100, the curves start to stabilize, showing a slight continuous decrease over subsequent epochs. Examining precision and recall (Figure 6c), it is observed that both parameters stabilize around epoch 206, with estimated values of 97.08% and 96.94%, respectively. Regarding mAP50 and mAP50-95 (Figure 6d), both metrics reach a plateau around epoch 170 with values of 97.92% and 87.9% successively.

Once the training is completed, the obtained weights are used for object detection on the live stream of RGB frames. The deployment of FastSAM is integrated, taking the detected bounding boxes as input for the FastSAM model. Upon detecting the mask, it is converted into a binary mask. The results of YOLO, FastSAM, and binary mask extraction are visualized in Figure 7.

Figure 7a illustrates the object detection results using the pre-trained YOLO model, highlighting its challenges in detecting certain objects (mainly the bottle) while achieving high accuracy for the other two objects cup and laptop (a mean value of 90%). Figure 7b, on the other hand, demonstrates that custom training is a solution for detecting specific objects not covered by pre-trained models, achieving a minimum accuracy of 91% for the bottle.

#### 4.3.2. Quantitative Analysis

A comprehensive evaluation of the trained object detection YOLO model has been conducted to assess the robustness and generalization capabilities across diverse environmental conditions. The evaluation process involves three distinct sets of images. Each set contains between 20 and 30 images, designed to represent different scenarios encountered in real-world deployment. (1) The first set of images comprises similar environmental and lighting conditions to those used during model training. These images serve as a baseline for assessing the model’s performance in familiar settings and providing insights into its ability to handle variations within its training domain. (2) The second set of images introduces variability in object positions, orientations, and lighting conditions compared to the training data. By capturing a broader range of scenarios, this set enables to evaluate the model’s adaptability to changes in object positions, orientations, and lighting, while simulating real-world challenges such as occlusions and shadows. (3) The third set of images presents a more significant departure from the training data by incorporating entirely different backgrounds, surfaces, and lighting conditions. This set aims to test the model’s generalization capabilities beyond its training domain, such as to assess its ability to detect objects accurately in novel environments with diverse visual characteristics.

Table 2 presents a comprehensive analysis of the YOLO model’s performance in terms of Intersection over Union (IoU) and precision metrics across the different test subsets.

Across these scenarios, the “cup” class consistently demonstrates superior performance, achieving high IoU and Precision scores across all test sets (0.96 and 0.99 for IoU and Precision, respectively). This performance suggests robustness in the model’s ability to accurately localize and classify instances of cups, regardless of environmental factors or object configurations. Conversely, the “bottle” class exhibits the lowest IoU and Precision scores, particularly for test set (3) with respective values of 0.52 and 0.31. It indicates additional challenges in accurately localizing and classifying bottle instances under more complex environmental conditions or object orientations.

In addition to YOLO evaluation, FastSAM has been analyzed through the annotation of one subset to create a set of ground truth instance segmentation masks. These masks have been overlapped and grouped in a single array, followed by a conversion to binary image, which allow an overall assessment of mask prediction quality. Afterwards, FastSAM has been applied to predict the objects masks by considering the predicted YOLO bounding boxes as inputs. The resulting mask is also converted to a binary image, and then compared to the ground truth one. The evaluation of segmentation algorithms involves assessing various metrics to gauge their performance. The Jaccard Index (also known as Intersection over Union) and Dice Coefficient [60] are key measures that evaluate the overlap between the predicted and ground truth masks, with higher values indicating better agreement. Precision quantifies the accuracy of positive predictions, while recall measures the ability to identify all relevant instances of the object [61]. The F1 Score balances precision and recall, offering a single metric that considers both false positives and false negatives. The Area under the ROC curve (AUC) assesses segmentation performance across different threshold settings by plotting the true positive rate against the false positive rate [62]. Pixel-wise Accuracy (PA) provides an overall measure of segmentation accuracy at the pixel level [63].

Upon evaluating a segmentation algorithm, the obtained results are summarized in Figure 8. The mean metrics demonstrate high values across various evaluation criteria: Jaccard Index (IoU) at 0.94, Dice Coefficient at 0.92, AUC at 0.95, Precision at 0.93, Recall at 0.94, F1 Score at 0.92, and Pixel Accuracy at 0.96. However, considering the standard deviation of the metrics helps in understanding the variability in the results. Despite generally favorable mean metrics, standard deviations shows some variability across evaluations (ranging from 0.12 for Pixel-wise Accuracy to 0.20) and indicates areas for potential improvement or optimization in the algorithm. Upon evaluating a segmentation algorithm, the obtained results are summarized in Figure 8. Since standard deviation analysis assumes a Gaussian distribution of the data, any disturbance (outliers due to inaccurate FastSAM mask estimation at certain sensor’s poses) can cause a misestimation (example in Figure 9). In such cases, the median absolute deviation values, ranging from 0.0029 to 0.0097, provide further insight into the spread of the data and complement the standard deviation analysis.

### 4.4. 3D Object Reconstruction and Discussion

The resulting mask is then aligned with depth frame using the default realsense parameters rs.align_to and K matrices [64]. The selected native resolution for both RGB and depth images are 640×480, which results into approximately 300k 3D points in the full-view reconstructed point-cloud. When applying the FusionVision pipeline, the background has been removed decreasing the number of points to around 32k and focusing the detection on the region of interest only, which leads to more accurate object identification.

Before performing 3D object reconstruction, the point-cloud undergoes downsampling and denoising procedures for enhanced visualization and accuracy. The downsampling is achieved using Open3D’s voxel-downsampling method with a voxel size of five units. Subsequently, statistical outlier removal is applied to the downsampled point-cloud with the following parameters: *neighbors* =300 and standard deviation *ratio*=2.0. These processes result in a refined and denoised point-cloud, addressing common issues such as noise and redundant data points. This refined point-cloud serves as the basis for precise 3D object reconstruction. The real-time performance of the YOLO and FastSAM has been approximated to $\frac{1}{30.6\;\mathrm{ms}}\approx 32.68\;\mathrm{fps}$, as the image processing involves three main components: preprocessing (1 ms), running the inference (27.3 ms), and post-processing the results (2.3 ms). When incorporating 3D processing and visualization of the raw, non-processed obtained 3D objects’ point-clouds, the real-time performance decreases to 5 fps. Thus, additional point-cloud post-processing is needed, including downsampling and denoising. The results are presented in Figure 10.

In Figure 10a, we can distinguish the presence of noises and wrong depth estimations, mainly due to the object reflectance and inaccurate calculation of disparity. Therefore, the post-processing increases the accuracy of 3D bounding box detection as shown in Figure 10b while maintaining an accurate representation of the 3D object.

The impact of different processing techniques on the distribution of points and object reconstructions derived from a raw point-cloud is illustrated in Figure 11: (a) raw point-cloud, (b) downsampled point-cloud, and (c) downsampled + denoised point-cloud:In Figure 11a, the raw point-cloud displays a relatively balanced distribution among different object categories. Notably, the computer and bottle categories contribute significantly, comprising 29.8% and 17.3% of the points, respectively. Meanwhile, the cup and other objects make up smaller proportions. This point-cloud presented several noise and inaccurate 3D estimation.In Figure 11b, where the raw point-cloud undergoes downsampling with voxel=5 without denoising, a substantial reduction in points assigned to the computer and bottle categories (4.7% and 2.3%, respectively) is observed, which improves the real-time performance while maintaining a good estimation of the object 3D structure.In Figure 11c, the downsampled point-cloud is further subjected to denoising. The distribution remains relatively similar to Figure 11b with a minor decreases in the computer and bottle categories (4.3% and 1.8%, respectively) while eliminating the point-cloud noise for each detected object.

Table 3 summarizes the frame rate evolution when applying the FusionVision Pipeline step by step.

The fusion of 2D image processing and 3D point-cloud data has led to a significant improvement in object detection and segmentation. By combining these two disparate sources of information, we have been able to eliminate over 85% of combined non-interesting and the noisy point-cloud, resulting in a highly accurate and focused representation of the objects within the scene. This allows the enhancement of scene understanding and enables reliable localization of individual objects, which can then be used as input for 6D object pose identification, 3D tracking, shape and volume estimation, and 3D object recognition. The accuracy and efficiency of the FusionVision pipeline make it particularly well-suited for real-time applications, such as autonomous driving, robotics, and augmented reality.

## 5. Conclusions

FusionVision stands as a comprehensive approach in the realm of 3D object detection, segmentation, and reconstruction. The outlined FusionVision pipeline encompasses a multi-step process, involving YOLO-based object detection, FastSAM model execution, and subsequent integration into the three-dimensional space using point-cloud processing techniques. This holistic approach not only amplifies the accuracy of object recognition, but also enriches the spatial understanding of the environment. The results obtained through experimentation and evaluation underscore the efficiency of the FusionVision framework. First, the YOLO model has been trained on a custom-created dataset then deployed on real-time RGB frames. FastSAM model has been subsequently applied on the frame while considering the detected objects bounding boxes to estimate their masks. Finally, point-cloud processing techniques have been added to the pipeline to enhance the 3D segmentation and scene understanding. This has led to the elimination of over 85% of unnecessary point-cloud for the 3D reconstruction of specific physical objects. The estimated 3D bounding boxes of the objects defines well the shape of the 3D object in the space. The proposed FusionVision method showcases high real-time performances particularly in indoor scenarios, which could be adopted in several applications, including robotics, augmented reality, and autonomous navigation. Through the deployment of FusionVision (NVIDIA GPU RTX 2080 Ti with 11 GB of memory), it allows reaching a real-time performance of about 27.3 fps (frames per second) while accurately reconstructing the objects in 3D from the RGB-D view. Such performance underscores the scalability and versatility of the proposed framework for real-world deployment. As perspectives, the continuous evolution of FusionVision could involve leveraging the latest zero-shot detectors to enhance its object recognition capabilities. Additionally, the investigation of Language Model (LLM) integration for operations such as prompt-based specific object identification and real-time 3D reconstruction stands as a promising avenue for future enhancements.

## Figures and Tables

**Figure 1 sensors-24-02889-f001:**
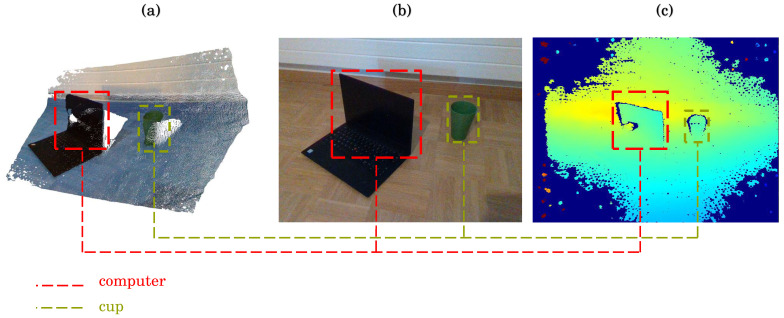
Example of RGB-D camera scene capturing and 3D reconstruction. (**a**) 3D reconstruction from RGB-D depth-channel. (**b**) RGB stream capture from RGB sensor. (**c**) Visual estimation of depth with the ColorMap JET (the closer object are represented in green and far ones are the dark blue regions).

**Figure 2 sensors-24-02889-f002:**
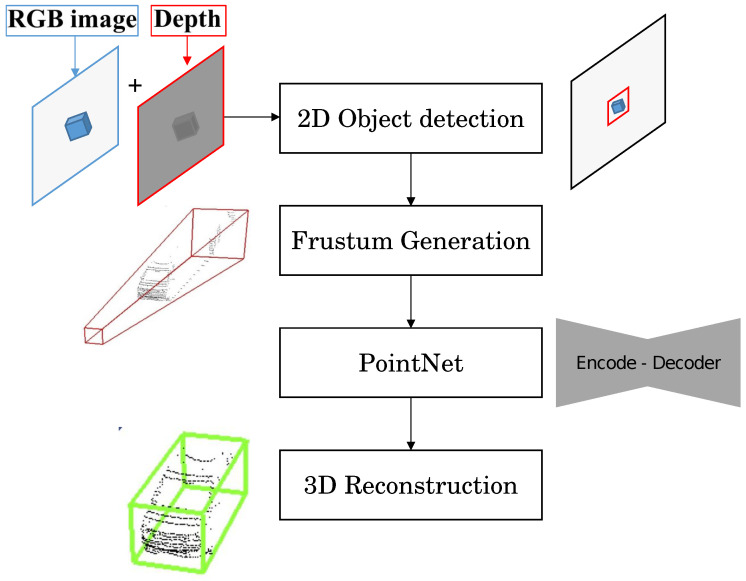
Complex YOLO framework for 3D object reconstruction and localization [47].

**Figure 3 sensors-24-02889-f003:**
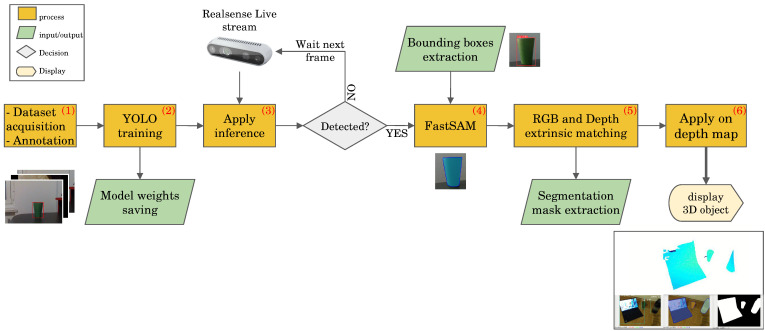
Proposed FusionVision pipeline for real-time 3D object segmentation and localization using fused YOLO and FastSAM applied on RGB-D sensor.

**Figure 4 sensors-24-02889-f004:**
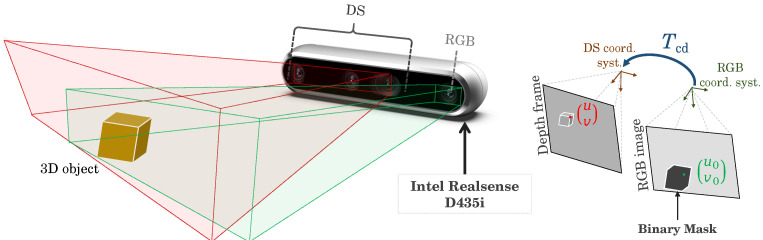
Visual representation of RGB camera alignment with the depth sensor.

**Figure 5 sensors-24-02889-f005:**
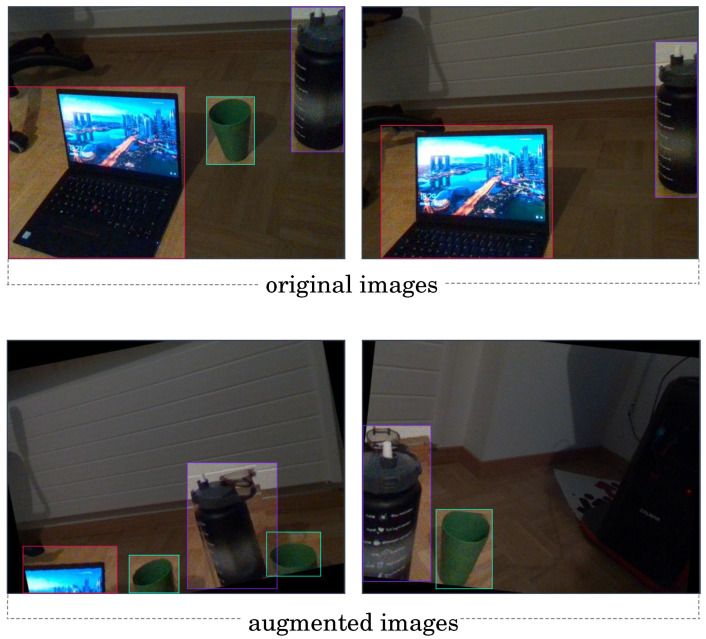
Example of acquired images for YOLO training: the top two images are original, the bottom ones are augmented images.

**Figure 6 sensors-24-02889-f006:**
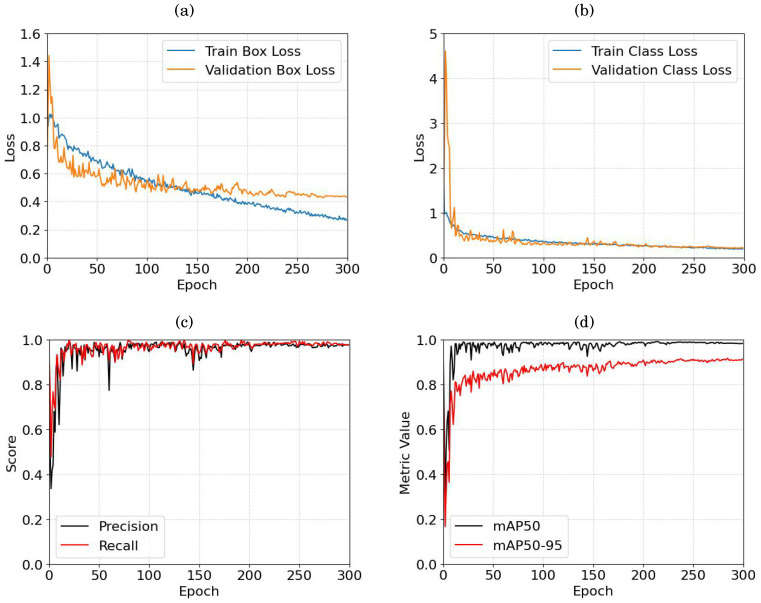
YOLO training curves: (**a**) bbox loss, (**b**) cls loss, (**c**) precision and recall, and (**d**) mAP50 and mAP50-95.

**Figure 7 sensors-24-02889-f007:**
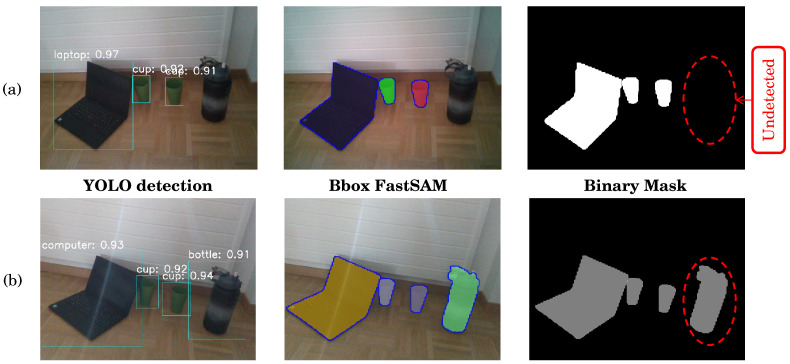
Visuals of the YOLO detection, FastSAM mask extraction, and binary mask estimation: (**a**) using the pre-trained YOLO model; (**b**) using the custom trained YOLO model.

**Figure 8 sensors-24-02889-f008:**
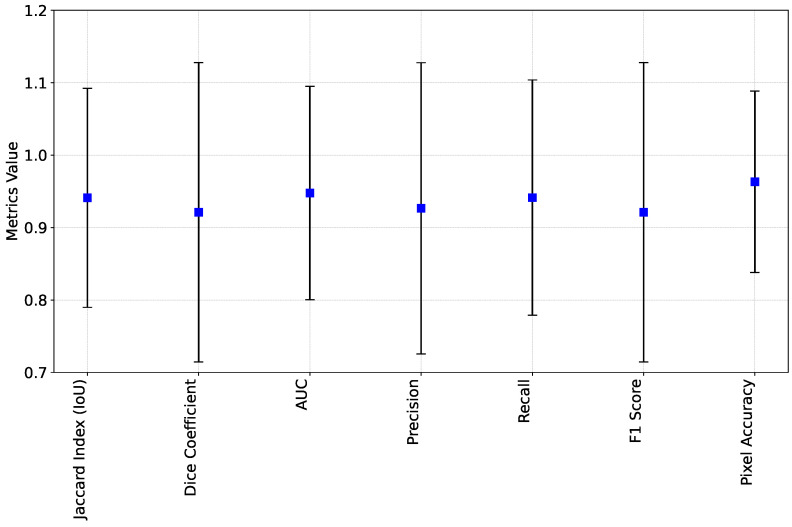
Overall evaluation metrics of FastSAM applied on extracted YOLO bounding boxes and compared to ground truth annotation. The blue points refers to the values of the metrics and black segments are standard deviations.

**Figure 9 sensors-24-02889-f009:**
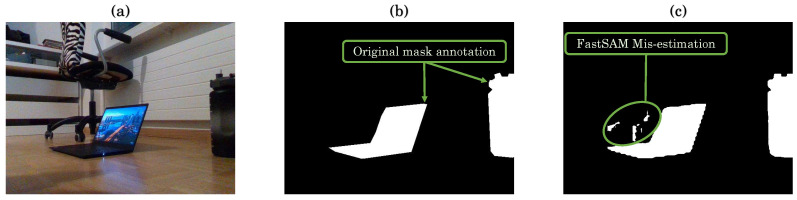
Example of FastSAM misestimation of the segmentation mask: (**a**) original image, (**b**) ground truth annotation mask, and (**c**) FastSAM estimated mask.

**Figure 10 sensors-24-02889-f010:**
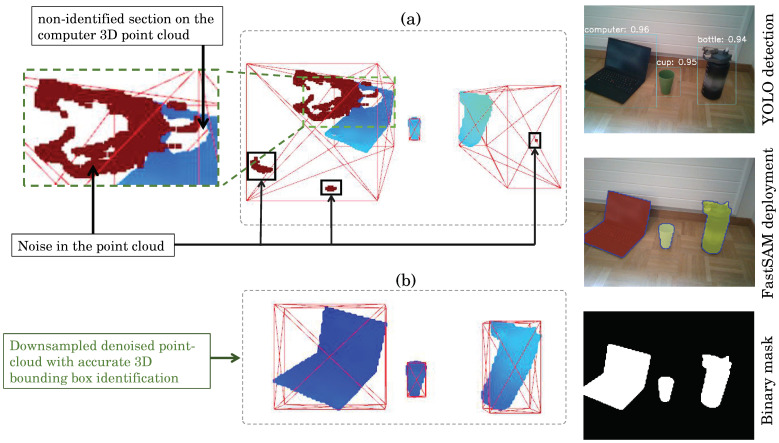
Three-dimensional object reconstruction from aligned FastSAM mask: (**a**) raw point-cloud and (**b**) post-processing point-cloud by voxel downsampling and statistical denoiser technique. The left images visualizing the YOLO detection, FastSAM mask extraction, and Binary mask estimation at specific positions of the physical objects within the frame.

**Figure 11 sensors-24-02889-f011:**
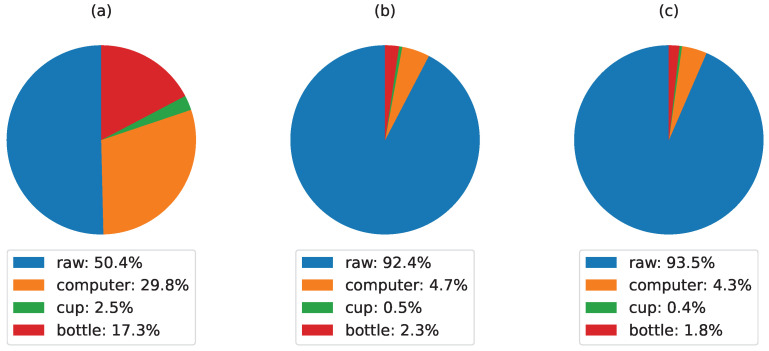
Post-processing impact on 3D object reconstruction: (**a**) raw point-clouds, (**b**) Downsampled point-clouds, and (**c**) Downsampled + denoised point-clouds.

**Table 1 sensors-24-02889-t001:** Setup configuration for realtime FusionVision pipeline.

Name	Version	Description
Linux	22.04 LTS	Operating system
Python	3.10	Baseline programming language
Camera	D435i	Intel RealSense RGB-D camera
GPU	RTX 2080 TI	GPU for data parallelization
OpenCV	3.10	Open source Framework for computer vision operations
CUDA	11.2	Platform for GPU based processing

**Table 2 sensors-24-02889-t002:** Summary of YOLO’s performance in bounding box estimation compared to ground-truth annotated three test subsets.

Metrics	Test Sets	Cup	Bottle	Computer	Overall
IoU	1	0.96	0.96	0.95	0.95
	2	0.93	0.90	0.91	0.92
	3	0.83	0.52	0.72	0.70
Precision	1	0.99	0.96	0.98	0.98
	2	0.91	0.77	0.85	0.87
	3	0.6	0.31	0.54	0.49

**Table 3 sensors-24-02889-t003:** Summary of frame rate improvement when applying FusionVision pipeline for 3D objects isolation and reconstruction.

Process	Processing Time (ms)	Frame Rate (fps)	Point-Cloud Density
Raw point-cloud visualization	∼16	up to 60	∼302.8 k
RGB + Depth map (Without point-cloud visualization)	∼11	up to 90	-
+ YOLO	∼31.7	∼34	-
+ FastSAM	∼29.7	∼33.7	-
+ Raw 3D Object visualization	∼189	∼5	∼158.4 k
complete FusionVision Pipeline	**∼30.6**	**∼27.3**	**∼20.8 k**

## Data Availability

The proposed FusionVision pipeline can be reproduced from the GitHub repository link https://github.com/safouaneelg/FusionVision (accessed on 28 February 2024). All the visualization, real-time performances, and 3D visualizations are within the repository.

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
