# Peer review of "FusionVision: A Comprehensive Approach of 3D Object Reconstruction and Segmentation from RGB-D Cameras Using YOLO and Fast Segment Anything"

_sensors, 2024, doi:10.3390/s24092889_

Round 1
Reviewer 1 Report
Comments and Suggestions for Authors
This paper presents a method for 3D object reconstruction and segmentation, which combines the YOLO object detection model and the FastSAM object segmentation model for RGB-D cameras to achieve better three-dimensional object detection through 3D point cloud reconstruction. It is interesting to integrate the latest object recognition and object segmentation models in RGB-D cameras, focusing the attention of object segmentation on the targets obtained from object recognition to reduce the computational cost. The post-processing involves down-sampling and denoising of the point cloud environment, leading to an enhancement in the accuracy of the entire system. This work exhibits a well-structured and logically sound approach. The proposed Fusion Pipeline effectively leverages the advantages of RGB-D cameras, achieving innovation in the field of 3D object reconstruction and segmentation, ultimately enhancing the accuracy of spatial object recognition. However, the paper could further improve and modify the presentation of the novelty and advantages of the proposed method.
1. In Figure 3, there might be a missing arrow pointing from the inference process to the process of determining whether the target is recognized; in the caption of Figure 9, there are two instances of (b).
2. The statement in lines 192 to 194 regarding focusing object segmentation on targets recognized by YOLO to reduce computational costs significantly lacks sufficient elaboration in the paper. It is recommended to supplement with comparisons to other similar methods or models.
3. The data presented in the conclusion section only demonstrate the improvement in object recognition accuracy due to point cloud post-processing; they lack a more detailed demonstration of the impact of point cloud post-processing on system real-time performance.
Comments on the Quality of English LanguageNULL
Author Response
Dear Reviewer,
We truly appreciate the time and effort you invested in carefully assessing our work and providing thoughtful recommendations for improvement. Your feedback has been instrumental in enhancing the rigor and clarity of our research paper. Additionally, supplementary analysis in experimental sections has been added to the paper in order to bring a better assessment of the proposed FusionVision Pipeline. Additional proposals have been treated and led to more changes in the text.
All your suggestions and comments have been carefully analyzed and answered in a separate file. All paper modifications have been highlighted in red text color and the paper has been merged to the letter document. Moreover, we have added a set of new elements including: Table 2 and 3, Figures 8 and 9.
You'll find attached to these notes the PDF file with comments answers and the paper with highlighted changes in red text color.
Best regards;
Dr. El Ghazouali
TOELT LLC

Reviewer 2 Report
Comments and Suggestions for Authors
The paper “FusionVision: A comprehensive approach of 3D Object reconstruction and segmentation from RGB-D cameras using YOLO and Fast Segment anything” needs some improvements.
· Please list your contributions to the paper.
· What is the problem statement you are going to address and what specific research gap you are going to cover?
· Please provide quantitative results of your research.
· Explain the architecture of your model with figures explicitly from pipeline.
· Perform quantitative and qualitative comparisons with other state-of-the-art models.
· Provide the results of your deployed model in a real time environment.
· Discuss why pretrained YOLO model ignores certain types of objects, explain its reasons.
· First discuss post-processing techniques then explain their impact on the results. Include the table, which includes results with and without post processing techniques.
Author Response
Dear Reviewer,
Thank you for the opportunity to revise our paper on the proposed FusionVision Pipeline. We appreciate the time and effort you have put into providing feedback, which has been invaluable in helping us improve our work. We took the time to answer, add and review our paper to make it as complete as possible while considering your recommendations.
We have carefully considered and replied to all the comments and made the necessary revisions. The changes are marked in red in the revised manuscript attached with the answer in the PDF. Below, we have provided a point-by-point response to your comments.
You'll find attached to these notes the PDF file with comments answers and the paper with highlighted changes in red text color.
Best regards;
Dr. El Ghazouali
TOELT LLC

Round 2
Reviewer 2 Report
Comments and Suggestions for Authors
You have mentioned the contributions and problem statement in the response file only. Please include it in the paper as well.
Author Response
Dr. Safouane El Ghazouali,
PostDoc researcher in Computer vision
TOELT LLC AI lab
safouane.elghazouali@toelt.ai
Dear Reviewer,
We would like to thank you for the second review round of our manuscript. Your constructive feedback has contributed to improve the quality and clarity of our research paper.
Please find below the answer and attached to this mail the letter and revised manuscript containing modification highlighted in red.
Thank you once again for your time and dedication to our manuscript.
Best regards,
Dr. El Ghazouali Safouane
PostDoc Researcher at TOELT LLC
Computer Vision and LLM
—---------------------------------------------------------------------------------------
Reviewer's comment. You have mentioned the contributions and problem statement in the response file only. Please include it in the paper as well.
ANSWER: Thank you for this observation. In response to your suggestions, we have diligently incorporated supplementary descriptions in regards to explicit listing of the contribution to the paper. The paragraph has been added in the introduction above the paper’s organizational structure and below Figure 1.
You will find the letter attached to this mail.
